# HP1γ Sensitizes Cervical Cancer Cells to Cisplatin through the Suppression of UBE2L3

**DOI:** 10.3390/ijms21175976

**Published:** 2020-08-19

**Authors:** Sang Ah Yi, Go Woon Kim, Jung Yoo, Jeung-Whan Han, So Hee Kwon

**Affiliations:** 1Epigenome Dynamics Control Research Center, School of Pharmacy, Sungkyunkwan University, Suwon 16419, Korea; angelna1023@hanmail.net; 2College of Pharmacy, Yonsei Institute of Pharmaceutical Sciences, Yonsei University, Incheon 21983, Korea; goun6997@daum.net (G.W.K.); jungy619@yonsei.ac.kr (J.Y.)

**Keywords:** cervical cancer, cisplatin resistance, HP1γ, UBE2L3, p53, leptomycin B, doxorubicin

## Abstract

Cisplatin is the most frequently used agent for chemotherapy against cervical cancer. However, recurrent use of cisplatin induces resistance, representing a major hurdle in the treatment of cervical cancer. Our previous study revealed that HP1γ suppresses UBE2L3, an E2 ubiquitin conjugating enzyme, thereby enhancing the stability of tumor suppressor p53 specifically in cervical cancer cells. As a follow-up study of our previous findings, here we have identified that the pharmacological substances, leptomycin B and doxorubicin, can improve the sensitivity of cervical cancer cells to cisplatin inducing HP1γ-mediated elevation of p53. Leptomycin B, which inhibits the nuclear export of HP1γ, increased cisplatin-dependent apoptosis induction by promoting the activation of p53 signaling. We also found that doxorubicin, which induces the DNA damage response, promotes HP1γ-mediated silencing of UBE2L3 and increases p53 stability. These effects resulted from the nuclear translocation and binding of HP1γ on the UBE2L3 promoter. Doxorubicin sensitized the cisplatin-resistant cervical cancer cells, enhancing their p53 levels and rate of apoptosis when administered together with cisplatin. Our findings reveal a therapeutic strategy to target a specific molecular pathway that contributes to p53 degradation for the treatment of patients with cervical cancer, particularly with cisplatin resistance.

## 1. Introduction

Cervical cancer is the fourth most frequently diagnosed cancer and the fourth common cause of mortality in women worldwide, with an estimated 570,000 cases and 311,000 deaths in 2018, despite the advancement of Papanicolaou smear screening and prophylactic vaccines [1]. In the early-stage of cervical cancer, surgery represents the primary treatment possibly associated with radiotherapy. However, in the advanced stages, radiochemotherapy or chemotherapy alone are the most appropriate choice although treatment options are very limited for recurred cervical cancer. [2]. The patients with advanced or recurrent cervical cancer have poor prognosis, with chemotherapy response rates ranging between 20% and 36% and a 1-year survival rate between only 10% and 20%. Historically, cisplatin, a chemotherapeutic platinum compound, was the most effective first-line therapy for cervical cancer [3]. Currently, patients with persistent, recurrent, or metastatic cervical cancer not amenable with surgery or radiotherapy are treated in combination with paclitaxel and either cisplatin or topotecan [4]. The combination treatment of those chemotherapy and bevacizumab, anti-angiogenesis drug, improves overall survival of advanced cervical cancer patients. [4]. The molecular mechanism underlying cisplatin-induced anticancer effects involves DNA damage response and apoptosis induction by causing DNA lesions [5]. When the cells are treated with cisplatin, the concentration of chloride ions decreases in the cytoplasm, and the chloride ligands of cisplatin are progressively substituted by water [6]. The highly reactive hydrated cisplatin covalently binds to the DNA and forms DNA–cisplatin adducts, resulting in the induction of DNA damage [5]. When cisplatin-induced DNA damage exceeds the DNA repair capacity, cisplatin exerts anticancer effects by inducing apoptosis [5].

However, resistance to cisplatin, whether intrinsic or acquired, may develop, and can severely reduce the efficacy of treating patients with advanced or recurrent cervical cancer [7]. Thus, the treatment of cisplatin-resistant cervical cancer continues to be a major challenge. Furthermore, increasing the cisplatin dose leads to further cytotoxicity in the bystander normal tissues [8]. Enhancing the efficacy of cisplatin in cisplatin-resistant cervical cancer requires therapeutic approaches that can potentiate cisplatin cytotoxicity and overcome resistance. The identification of the molecular mechanisms that can regulate cisplatin resistance and can be exploited to develop novel strategies for the treatment of cisplatin-resistant cervical cancer are the key to improve the survival rate of the patients with advanced or recurrent cervical cancer.

Patients with cervical cancer primarily carry a wild-type *TP53* gene, and cervical carcinogenesis is highly correlated with persistent high-risk human papillomavirus (HPV) infection [9]. The high-risk HPV E6 viral oncoprotein is critical for inducing malignant transformations because it inhibits the tumor suppressor functions of p53 [9]. The E6 protein causes rapid degradation of p53 [10], which is dependent on E3 ubiquitin ligase, E6AP, and E2 ubiquitin conjugating enzyme UBE2L3 [11,12], subsequently suppressing p53-induced growth arrest and apoptosis. Mutations in the *TP53* gene are known to be associated with cisplatin resistance [13]. The reintroduction of p53 increases cisplatin sensitivity in *TP53* null ovarian cancer cells and enhances cisplatin-induced apoptosis. Notably, our previous findings revealed that HP1γ, a reader protein of methylated histone H3 lysine 9 (H3K9me) [14], can recover the p53 protein expression levels through the epigenetic silencing of UBE2L3 [12]. We also found that HPV E6 expression leads to abnormal nuclear export of HP1γ, which contributes to excessive ubiquitination of p53 by UBE2L3 [15]. Thus, we strived to identify a way to overcome cisplatin resistance by utilizing HP1γ-mediated p53 stability.

Here, we identified the pharmacological strategies to sensitize cervical cancer cells to cisplatin by inducing HP1γ-mediated p53 elevation. For these purposes, we evaluated the effects of leptomycin B (LMB), an exportin-1 inhibitor, and doxorubicin, which induces DNA damage, on cell proliferation, apoptosis, and reactivity to cisplatin. It has been reported that both LMB and doxorubicin promotes initial activation of p53 and accumulation of p53 in the nucleus, respectively [16]. In this study, we discovered the mechanism underlying p53 activation upon LMB or doxorubicin. LMB increased the cisplatin-dependent apoptosis by suppressing UBE2L3 and promoting p53 signaling in cisplatin-resistant cervical cancer cells. Moreover, doxorubicin treatment increased the nuclear retention and UBE2L3 promoter enrichment of HP1γ, subsequently promoting HP1γ-mediated silencing of UBE2L3 and increasing the p53 stability and activity in cisplatin-resistant cervical cancer cells. Our results suggest that the inhibition of nuclear export of HP1γ facilitates overcoming the cisplatin resistance caused by HPV-mediated loss of p53 in cervical cancer cells.

## 2. Results

### 2.1. Overexpression of HP1γ Increases the Sensitivity of Cervical Cancer Cells to Cisplatin

To assess the involvement of HP1γ in cisplatin response, we treated cisplatin-sensitive and cisplatin-resistant cervical cancer cell lines [17], with 5 μM cisplatin after ectopically overexpressing HP1γ. The cytotoxic effects of cisplatin were observed in the cisplatin treated HeLa cells, whereas SiHa and CaSki cells, which are more resistant to cisplatin than HeLa cells [17,18], did not show any morphological changes (Figure 1A). However, overexpression of HP1γ decreased the cell viability (Figure 1A), largely reducing the IC_50_ values of cisplatin in the three cervical cancer cell lines (107 to 7.33 nM in HeLa cells, 12.7 to 0.9 μM in SiHa cells, 3.73 to 0.06 μM in CaSki cells) (Figure 1B). It has been reported that p53/p21 signal is more prominent in cisplatin-sensitive HeLa cells than cisplatin-resistant SiHa and CaSki cells [17]. As our previous study showed that overexpression of HP1γ induces apoptosis in cervical cancer cells by inhibiting the UBE2L3-mediated p53 degradation [15], we examined the involvement of HP1γ-UBE2L3-p53 axis in sensitizing cervical cancer cells to cisplatin. In HeLa cells, which were observed to be most sensitive to cisplatin among the three cell lines (Figure 1A,B), cisplatin treatment increased the expression levels of p53 and Puma, a proapoptotic marker induced by p53 (Figure 1C, left). However, in SiHa and CaSki cells, cisplatin treatment did not increase the expression levels of p53 and Puma as much as in HeLa cells (Figure 1C, middle and right). Overexpression of HP1γ reduced the expression of UBE2L3 and increased the protein levels of p53 and Puma in the three cervical cancer cell lines (Figure 1C). These results suggest that overexpression of HP1γ sensitizes cervical cancer cells to cisplatin and increases p53 signaling even in cisplatin-resistant cells.

### 2.2. Inhibiting the Nuclear Export of HP1γ Increases the Sensitivity of Cervical Cancer Cells to Cisplatin

We previously revealed that the treatment of cervical cancer cells with LMB, an exportin-1 inhibitor, prevents the nuclear export of HP1γ, and thus, suppresses the expression of UBE2L3 while promoting p53 signaling [15]. Therefore, we examined whether co-treatment with LMB and cisplatin can sensitize cisplatin resistant (SiHa) cells. Treatment of SiHa cells with LMB increased the levels of p53 and its target genes, but decreased the expression of UBE2L3 both in the absence and presence of cisplatin (Figure 2A,B). In line with the synergistic increase in p53 target gene transcription (Figure 2B), the combined treatment of SiHa cells with cisplatin and LMB significantly increased the apoptotic cell population, as measured by flow cytometry (Figure 2C), and synergistically decreasing the cell proliferation rate (Figure 2D). Treatment of SiHa cells with LMB reduced the IC_50_ values of cisplatin by more than 10-fold (Figure 2E), substantiating the synergistic effects of LMB and cisplatin. These data indicate that the combination treatment with agents that prevent nuclear export of HP1γ can effectively sensitize cisplatin-resistant cervical cancer cells.

### 2.3. Doxorubicin-Mediated Elevation of p53 in Cervical Cancer Cells Requires the Suppression of UBE2L3 by HP1γ

We assessed the effects of another frequently used anti-cancer drug, doxorubicin, on cervical cancer cells. Treatment of cancer cells with doxorubicin led to apoptosis induction via the activation of the p53 signaling pathway [19]. Thus, we investigated whether HP1γ plays a role in doxorubicin-mediated induction of p53 expression. Interestingly, the treatment of doxorubicin in the three cervical cancer cells reduced the expression of UBE2L3 and increased the protein levels of p53, which were both reversed by HP1γ knockdown (Figure 3A–D). These effects resulted from the changes in p53 protein stability (Figure 3E) without altering the mRNA levels of p53 (Figure 3F). Accordingly, the transcription of p53 target genes was elevated by doxorubicin treatment but recovered by HP1γ knockdown (Figure 3G).

On the contrary, overexpression of HP1γ boosted the effects of doxorubicin on the downregulation of both the protein and mRNA levels of UBE2L3 (Figure 4A,B). Overexpression of HP1γ also enhanced the doxorubicin-mediated increase in the p53 protein levels (Figure 4A) by suppressing the ubiquitination of p53 (Figure 4C), but did not alter the mRNA levels of p53 (Figure 4D). The expression of p53 target genes was also significantly increased by overexpression of HP1γ compared with the doxorubicin-only treated group (Figure 4E). Taken together, our results suggest that doxorubicin treatment suppresses the expression of UBE2L3 and elevates the p53 signaling in cervical cancer cells, and HP1γ is required for these effects of doxorubicin.

### 2.4. Doxorubicin Treatment Increases the Nuclear Retention and UBE2L3 Promoter Enrichment of HP1γ

To better understand the mechanism of UBE2L3 suppression by doxorubicin, the localization of HP1γ was analyzed after treating cervical cancer cells with doxorubicin. Western blot analysis showed that the treatment of HeLa cells with doxorubicin increased the expression levels of nuclear HP1γ and decreased the cytoplasmic HP1γ (Figure 5A), implying nuclear import of HP1γ upon doxorubicin treatment. The chromatin immunoprecipitation (ChIP) assay data showed that doxorubicin treatment promoted the binding of HP1γ to the promoter region of *UBE2L3* gene in cervical cancer cells (Figure 5B–D). As HP1γ is a reader protein of repressive histone mark, H3K9me3, we further measured the enrichment of H3K9me3 and binding of H3K9 methyltransferases on the *UBE2L3* promoter. Similar to the binding of HP1γ, the enrichment of H3K9me3 and H3K9 methyltransferases, G9a and Suv39h1, on the *UBE2L3* promoter was remarkably enhanced by doxorubicin treatment (Figure 5B–D). Considering that H3K9me3 and HP1 encourage the formation of transcriptionally silent heterochromatin [20], doxorubicin-mediated binding of HP1 and H3K9 methyltransferases may contribute to heterochromatinization of the *UBE2L3* promoter region suppressing its transcription.

### 2.5. Doxorubicin Treatment Increases the Sensitivity of Cervical Cancer Cells to Cisplatin

We next evaluated whether doxorubicin, which mediates the nuclear import of HP1γ and UBE2L3 suppression, can reverse cisplatin resistance in SiHa cells that showed the highest IC_50_ value of cisplatin among the three cervical cancer cell lines (Figure 1B). The expression of UBE2L3 declined after dual treatment with doxorubicin and cisplatin, as compared with that observed with doxorubicin or cisplatin treatment alone (Figure 6A). Accordingly, the levels of p53 and apoptosis inducers (Bax and Puma) were much more elevated in the co-treated cells than in the cells treated with doxorubicin or cisplatin alone (Figure 6A). Consistent with the induction of p53 expression levels, the apoptotic cell population was dramatically increased upon the combination treatment (Figure 6B). Lastly, the IC_50_ value of SiHa cells for cisplatin was markedly decreased by co-treatment with doxorubicin and cisplatin (Figure 6C). These data demonstrate that the sensitivity of cisplatin-resistant cervical cancer cells can be restored by doxorubicin treatment.

## 3. Discussion

Cisplatin, one of the small molecule platinum compounds, is a widely used chemotherapeutic agent to treat cancer. Thus, extensive studies have been conducted to demonstrate the molecular mechanisms that can render cancer cells less susceptible to cisplatin [5]. The mechanisms underlying the resistance to cisplatin are complicated and generally related to the following characteristics: (1) reduced intracellular accumulation of cisplatin due to its decreased uptake and increased efflux; (2) increased DNA damage tolerance and repair; (3) suppression of apoptosis; (4) activation of epithelial-mesenchymal transition; and (5) alterations in the DNA methylation and microRNA profiles, expression of antioxidant proteins and stress-response chaperones, and acquisition of the cancer stem cell-like features [6]. Of these features, the inactivation of cisplatin-induced apoptosis is essential for the decline in the anticancer effects and the increase in chemoresistance against cisplatin. Cisplatin induces apoptosis by modulating multiple apoptotic proteins, such as p53 and Bcl-2 family proteins and cell survival signaling pathways, such as MAPK and NF-κB pathways [6]. For example, the activation and stabilization of the wild-type p53 are pivotal for cisplatin-induced apoptosis. Thus, the deficiency of p53 diminishes apoptosis and results in tolerance for DNA damage, consequently boosting drug resistance [21]. Cisplatin-based chemotherapy is more beneficial for cervical cancer patients with wild-type p53 than for those with mutated p53 [22]. Indeed, the percentage of wild-type p53 harboring cervical cancer cells is higher in the responders to cisplatin than in the nonresponders [23]. Thus, the upregulation or rescue of wild-type p53 can be a potential strategy to overcome the chemoresistance in cervical cancer, and p53 can be used as its predictive factor [24]. Although it was reported that ectopic expression of high-risk HPV E6 sensitized cervical cancer cells to cisplatin [25], a majority of studies have demonstrated that suppression of HPV E6 enhances the sensitivity to cisplatin [26,27,28]. Indeed, high-risk HPV harnesses a distinct mechanism to eliminate the protective effects of p53 in cervical cancer. The E6 protein derived from high-risk HPV stimulates poly-ubiquitination of p53 by E6AP (E3 ligase) and UBE2L3 (E2) [11,12]. Previously, we found a way to block the cervical cancer-specific p53 degradation by inducing HP1γ-mediated suppression of UBE2L3 [15]. Moreover, we revealed that E6 promoted cytoplasmic diffusion of HP1γ, disturbing the epigenetic silencing of UBE2L3 [15]. In the current study, we further demonstrate a pharmacological strategy targeting these mechanisms to resolve cisplatin resistance in cervical cancer cells. Doxorubicin and leptomycin B inhibited the nuclear export of HP1γ and suppressed the expression of UBE2L3, which can enhance the p53 stability (Figure 6D). Both agents successfully sensitized the cisplatin-resistant cervical cancer cells. These results prove that the strategies targeting UBE2L3-mediated p53 degradation can be also feasible for resolving cisplatin resistance in cervical cancer cells.

Exportin-1, a transporter protein, is responsible for shuttling of many tumor suppressors from the nucleus to the cytoplasm [29,30]. The dysregulation of exportin-1 has been demonstrated to promote tumorigenesis through excessive transport of the tumor suppressors, such as Rb, p53, and p21, to the cytoplasmic compartment [31]. As a result, selective inhibitors of nuclear export (SINE) have been developed as a promising class of anti-cancer agents [31,32]. Among the small molecules that inhibit exportin-1 function, selinexor (KPT-330) is under clinical trials for most cancer types, including leukemia, and lung, prostate, gastric, breast, ovarian, and cervical cancers [32]. Although the synergistic effects of the combination of selinexor and cisplatin have been evaluated in the ovarian cancer preclinical models [33], but in cervical cancer, selinexor is only been reported to be effective as a single-agent [34]. Herein, we suggest for the first time that exportin-1 inhibitor can be used in combination with cisplatin to ameliorate cervical cancer progression. The abnormal upregulation of exportin-1 and nuclear export of p53 by exportin-1 have been observed in cervical cancer specimen [35,36]. In addition to these prior studies, we showed the aberrant nuclear export of HP1γ by exportin-1 in cervical cancer cells and tissues [15]. The treatment of cervical cancer cells with LMB inhibited the interaction between exportin-1 and HP1γ, leading to HP1γ-mediated suppression of UBE2L3 and elevation of p53 [15]. Hence, our current results show that the synergistic effects of LMB and cisplatin apparently also stem from the nuclear retention of HP1γ and upregulation of functional p53. Further clinical or pre-clinical studies are required to apply this approach for inhibiting exportin-1 in cervical cancer in combination with cisplatin.

Another notable issue in this study is that we have elucidated a novel action of doxorubicin in controlling the subcellular localization of HP1γ and its enrichment on the target gene promoter. There have been two canonical mechanisms underlying the anticancer effects of doxorubicin: (1) intercalation into DNA and disruption of topoisomerase II-mediated DNA repair; (2) oxidative damage to the proteins and DNA by generating free radicals [37]. Our current findings demonstrate that doxorubicin plays a precise function at the transcriptional level. Upon doxorubicin treatment, the expression levels of HP1γ in the nucleus and its local levels on the UBE2L3 promoter were increased. Interestingly, H3K9me3 levels and binding of H3K9 methyltransferases (G9a and Suv39h1) on the UBE2L3 promoter were also significantly enhanced. These results also a raise question as to whether doxorubicin can affect other genes that are under the control of HP1γ and H3K9me3. A recent study showed a genome-wide transcriptome response to the long-term treatment with doxorubicin [38], but the mechanism of doxorubicin-mediated extensive transcriptomic alterations is still undefined. Future work might possibly uncover the epigenetic influence of doxorubicin on diverse cell biology, especially on H3K9me3 and HP1γ. Nevertheless, HP1γ knockdown did not fully recover UBE2L3 suppression by doxorubicin treatment (Figure 3A,C) indicating that doxorubicin affects UBE2L3 expression through a mechanism other than HP1γ. Considering that the doxorubicin-mediated decrease in *UBE2L3* mRNA level was completely restored by si-HP1γ (Figure 3D), doxorubicin might regulate stability or translation of UBE2L3 protein without being affected by HP1γ.

Although many strategies using combinatory regimens have been attempted to bypass cisplatin resistance, most of them failed to promote the therapeutic profile of cisplatin in clinical trials [39]. Here, we took advantage of targeting the HPV-mediated HP1γ export and UBE2L3-mediated p53 degradation, two distinct pathologies of cervical cancer. The clinical application of our approach may improve the outcome of cervical cancer patients with cisplatin resistance.

## 4. Materials and Methods

### 4.1. Cells Lines and Treatment

HeLa (ATCC^®^ CCL-2), SiHa (ATCC^®^ HTB-35), and CaSki (ATCC^®^ CRL-1550) cells were purchased from the American Type Culture Collection (ATCC, Manassas, VA, USA) and cultured according to the instructions from ATCC. The cells were maintained under a fully humidified atmosphere of 95% air and 5% CO_2_ at 37 °C. For treatment with agents, the cells grown to 80–90% confluence were treated with 50 nM of leptomycin B (Sigma-Aldrich, L2913, St. Louis, MO, USA), 5 μM of cisplatin (Sigma-Aldrich, 479306), or 1 μg/mL of doxorubicin (Sigma-Aldrich, D1515) for the indicated times.

### 4.2. Overexpression of HP1γ

For the ectopic expression of HP1γ, the cells were transfected with pcDNA-EGFP-HP1γ using Lipofectamine 2000 reagent (Life Technologies, Carlsbad, CA, USA) according to the manufacturer’s protocol. After 24–36 h, the proteins or RNAs were extracted from the cells.

### 4.3. Knockdown of HP1γ

For the knockdown of HP1γ, the cells were transfected with siRNA targeting HP1γ using Lipofectamine 2000 reagent (Life Technologies) according to the manufacturer’s protocol. After 48–72 h, the proteins or RNAs were extracted from the cells. The siRNA sequences targeting HP1γ are as follow: sense, 5′-AUUCUUCAGGCUCUGCCUC-3′ and antisense, 5′-GAGGCAGAGCCUGAAGAAU-3′.

### 4.4. Cell Viability and IC_50_ Values

The cells were seeded in 6-well plates (10^5^ cells/well). Twenty-four hours after the seeding, the cells treated with chemicals and then the cell number was counted every 8 h. For the cell number counting, the cells were detached from the plate with EDTA solution. The detached cells were washed with PBS, and the cell number was counted using LUNA-II™ Automated Cell Counter (Logos Biosystems). IC_50_ values of cisplatin in the cells were calculated from the viability graph with GraphPad Prism 8 (*n* = 3).

### 4.5. Flow Cytometry

Apoptosis was analyzed with Annexin V Apoptosis Detection Kit (BD Biosciences, Franklin Lakes, NJ, USA) following the manufacturer’s protocol (Annexin V-FITC Apoptosis Detection Kit; BD556547, BD Pharmingen, San Diego, CA, USA). SiHa cells were treated with the drugs and incubated for 48 h. The concentrations of the drugs used for the apoptosis assay were 5 µM cisplatin and 1 µg/mL doxorubicin. Next, SiHa cells were stained with Annexin V-FITC and propidium iodide in 1 × Annexin V binding buffer for 15 min. DMSO was used for negative control while doxorubicin and leptomycin B served as positive controls for each experiment. The apoptosis induction was detected using flow cytometer and BD FACSDiva software version 7 (both from BD Biosciences, Franklin Lakes, NJ, USA).

### 4.6. Immunoblotting

The cells were lysed in Pro-Prep (iNtRON Biotechnology, Seongnam, Korea) and the lysates were centrifuged at 13,000 rpm at 4 °C for 18 min. For immunoblotting, the proteins of each sample were separated by SDS-polyacrylamide gel electrophoresis (PAGE). The proteins were transferred to polyvinylidene difluoride (PVDF) membranes using a semi-dry transfer apparatus (Bio-Rad, Hercules, CA, USA). The membranes were incubated overnight with the indicated primary antibodies, and then incubated with horseradish peroxidase-conjugated secondary antibodies for 1 h (Abcam, Cambridge, UK). The signals were detected using the chemiluminescence reagents (AbClon, Guro, Republic of Korea) and quantified with ImageJ. Actin was used as an equal loading control, and the relative intensity of each band was calculated setting control levels as 1. For the primary antibodies, anti-p53 (Santa Cruz Biotechnology, SC-126, Dallas, TX, USA), anti-UBE2L3 (Abcam, ab108936, Cambridge, UK), anti-GFP (Santa Cruz Biotechnology, SC-9996), anti-actin (Millipore, mab1501), anti-Bax (Santa Cruz Biotechnology, SC-20067), anti-Noxa (Santa Cruz Biotechnology, SC-56169), anti-Puma (Santa Cruz Biotechnology, SC-28226), anti-Caspase 3 (Abcam, ab13585-100), anti-HP1γ (Millipore, 05-690, Burlington, MA, USA), anti-ubiquitin (Santa Cruz Biotechnology, SC-9133), anti-α-tubulin (Santa Cruz Biotechnology, SC-32293), and anti-Lamin A/C (Cell Signaling Technology, #2032) were used in this study.

### 4.7. Immunoprecipitation

The cells were lysed in Pro-Prep (iNtRON Biotechnology) for 20 min on ice and then centrifuged at 13,000 rpm at 4 °C for 20 min. The supernatants were incubated with the primary antibody against p53 (Santa Cruz Biotechnology, SC-126), overnight at 4 °C, followed by incubation with anti-rabbit Ig-IP beads (Rockland Immunochemicals, TrueBlot, Limerick, PA, USA) for 1 h at 4 °C. The beads were spun down at 2000 rpm for 1 min and washed three times with PBS. The proteins were eluted from the beads by boiling for 5 min in 2 × Laemmli buffer (Bio-Rad) and subjected to immunoblotting.

### 4.8. Nuclear Fractionation

Fractionation of the cells to separate the subcellular parts was performed as previously described [15]. The cells were lysed in Buffer A (10 mM HEPES containing 1.5 mM MgCl_2_, 10 mM KCl, 1 mM EDTA, 1 mM DTT, 0.5 μg/mL leupeptin, 1 mM PMSF, 1 μM pepstatin A, and 0.05% NP-40), and centrifuged at 3000 rpm at 4 °C for 10 min. The supernatants were separated as the cytoplasmic extracts and the pellets were resuspended in Buffer B (20 mM HEPES containing 1.5 mM MgCl_2_, 420 mM KCl, 25% glycerol, 0.2 mM EDTA, 1 mM DTT, 0.5 μg/mL leupeptin, 1 mM PMSF, and 1 μM pepstatin A) and incubated on ice for 30 min. The nuclear extracts were obtained by centrifugation at 13,000 rpm at 4 °C for 20 min. The cytoplasmic and nuclear extracts were subjected to immunoblotting.

### 4.9. Reverse Transcription and Quantitative Real-Time PCR (RT-qPCR)

Total RNA was extracted from the cells utilizing Easy-Blue reagent (Intron Biotechnology) according to the manufacturer’s instructions. Thereafter, 1 μg of the total RNA was reverse transcribed into cDNA using Maxim RT-PreMix Kit (Intron Biotechnology). Quantitative real-time PCR (qPCR) was performed by mixing cDNA, KAPA SYBR^®^ FAST qPCR Master Mix (Kapa Biosystems, Wilmington, MA, USA), and each primer below. The qPCR reaction was detected by a CFX96 Touch or Chromo4 real-time PCR detector (Bio-Rad). The relative mRNA levels were normalized to the levels of β-actin mRNA for each reaction. The qPCR primer sequences used are listed in Table 1.

### 4.10. Chromatin Immunoprecipitation and qPCR (ChIP-qPCR)

Chromatin immunoprecipitation was performed as previously described [40]. Briefly, the cells were incubated with 1% formaldehyde for cross-linking, followed by shearing. The chromatin solution was obtained by centrifugation at 13,000 rpm at 4 °C rpm for 20 min. A small portion (5%) of the chromatin solution was reserved as input DNA, and the remaining solution was incubated with the primary antibodies and protein A agarose/salmon sperm (Millipore, #16-157) overnight at 4 °C. Next, the chromatin fragments were de-crossed from the proteins and eluted to be subjected to qPCR using primers listed in Table 2. Anti-HP1γ (Millipore, 05-690), anti-trimethyl Histone H3 Lys9 (Millipore, 07-442), anti-G9a (Abcam, ab40542), and anti-Suv39h1 (Abcam, ab12405) were used for ChIP.

### 4.11. Statistical Analysis

Statistical significance was analyzed by two-tailed Student’s t-test with excel and assessed based on the *p*-value. Data represent the mean ± SEM for *n* = 3. * *p* < 0.05, ** *p* < 0.01, *** *p* < 0.001 vs. the control group; # *p* < 0.05, ## *p* < 0.01, and ### *p* < 0.001 vs. single agent-treated group.

## 5. Conclusions

In this study, we have elucidated that leptomycin B and doxorubicin induce HP1γ-mediated p53 stabilization and enhance the cisplatin sensitivity of cisplatin-resistant cervical cancer cells. These findings provide a novel promising therapeutic strategy that can target p53 degradation to overcome cisplatin resistance and achieve better therapeutic effects in patients with cervical cancer.

## Figures and Tables

**Figure 1 ijms-21-05976-f001:**
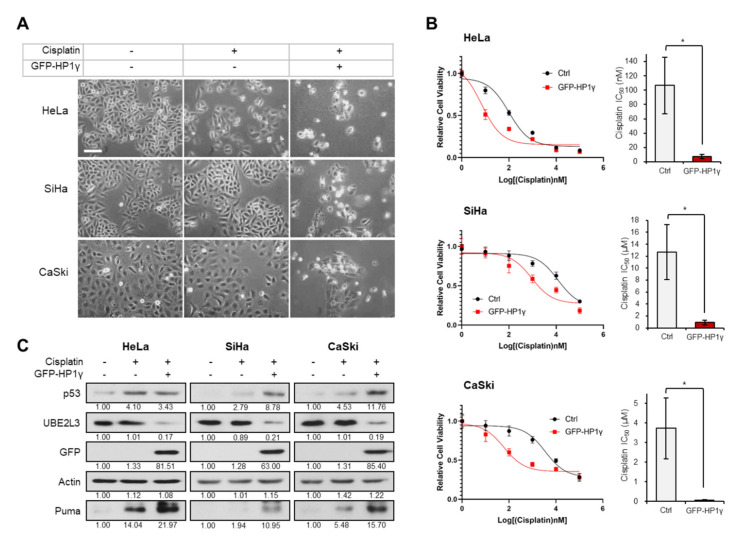
Overexpression of HP1γ increases the response of cervical cancer cells to cisplatin. (**A**) Microscopic images of GFP-HP1γ-expressing cervical cancer cells (HeLa, SiHa, and CaSki) treated with or without cisplatin (5 μM, 24 h). Scale bar, 100 μm. (**B**) Viability of HeLa, SiHa, and CaSki cells was measured 24 h after transfection with GFP-HP1γ. IC_50_ values of cisplatin in three cells transfected with GFP-HP1γ were calculated from the viability graph with GraphPad Prism 8 (*n* = 3). (**C**) Immunoblotting analysis of GFP-HP1γ-expressing cervical cancer cells (HeLa, SiHa, and CaSki) treated with or without cisplatin (5 μM, 24 h). Data are presented as the mean ± SEM (*n* = 3). * *p* < 0.05; vs. the control group.

**Figure 2 ijms-21-05976-f002:**
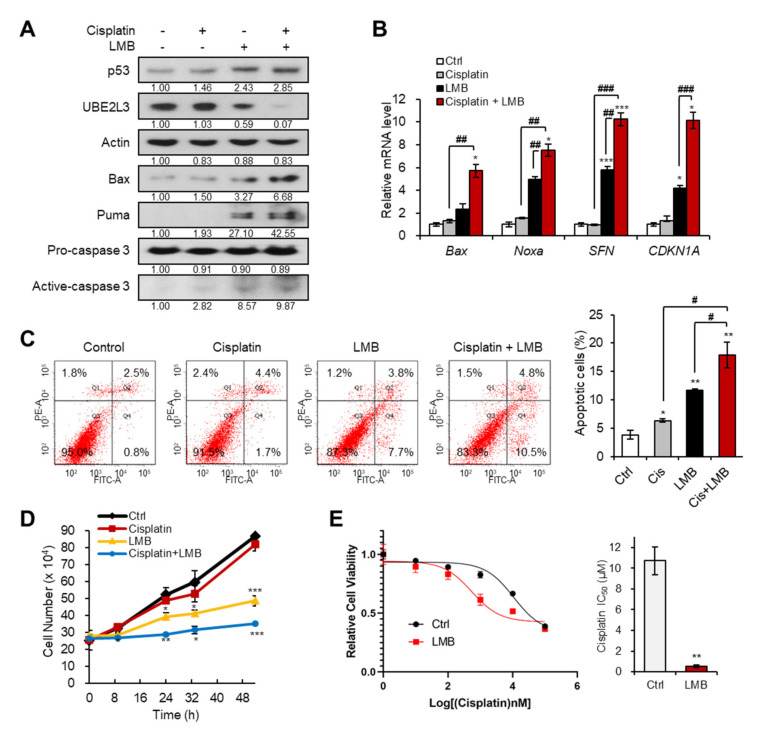
Exportin-1 inhibitor increases the response of cervical cancer cells to cisplatin. (**A**) Immunoblotting analysis of SiHa cells treated with or without cisplatin (5 μM) and LMB (50 nM) for 24 h. (**B**) The mRNA levels of p53 target genes (*Bax*, *Noxa*, *SFN*, and *CDKN1A*) in SiHa cells treated with or without cisplatin (5 μM) and LMB (50 nM) for 24 h were obtained with flow cytometry analysis. (**C**) Apoptotic cell populations of SiHa cells treated with or without cisplatin (5 μM) and LMB (50 nM) for 24 h were obtained with flow cytometry analysis. (**D**) The growth rate of SiHa cells treated with or without cisplatin (5 μM) and LMB (50 nM). *p*-values between control and samples were calculated and marked. (**E**) Viability and IC_50_ values of cisplatin in SiHa cells treated with LMB (50 nM) for 24 h. IC_50_ values of cisplatin in the cells were calculated from the viability graph with GraphPad Prism 8 (*n* = 3). Data are presented as the mean ± SEM (*n* = 3). * *p* < 0.05; ** *p* < 0.01; *** *p* < 0.001 vs. the control group; # *p* < 0.05, ## *p* < 0.01 and ### *p* < 0.001 vs. single agent-treated group.

**Figure 3 ijms-21-05976-f003:**
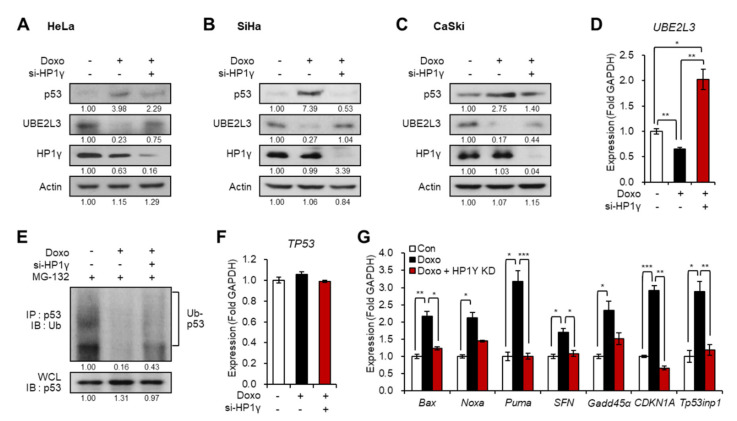
Suppression of UBE2L3 by HP1γ is required for doxorubicin-mediated p53 elevation in cervical cancer cells. (**A**–**C**) Immunoblot analysis of HeLa (**A**), SiHa (**B**), and CaSki (**C**) cells expressing siRNA of HP1γ in the presence of doxorubicin (1 μg/mL, 1 h). (**D**) The mRNA levels of *UBE2L3* gene in HeLa cells expressing siRNA of HP1γ in the presence of doxorubicin (1 μg/mL, 1 h). (**E**) Immunoblot analysis of p53 immunoprecipitates (IP) and whole cell lysates (WCL) from HeLa cells expressing siRNA of HP1γ in the presence of doxorubicin (1 μg/mL, 1 h) and MG-132. (**F**,**G**) The mRNA levels of *TP53* gene (**F**) or p53 target genes (**G**) in HeLa cells expressing siRNA of HP1γ in the presence of doxorubicin (1 μg/mL, 1 h). Data are presented as the mean ± SEM (*n* = 3). * *p* < 0.05; ** *p* < 0.01; *** *p* < 0.001.

**Figure 4 ijms-21-05976-f004:**
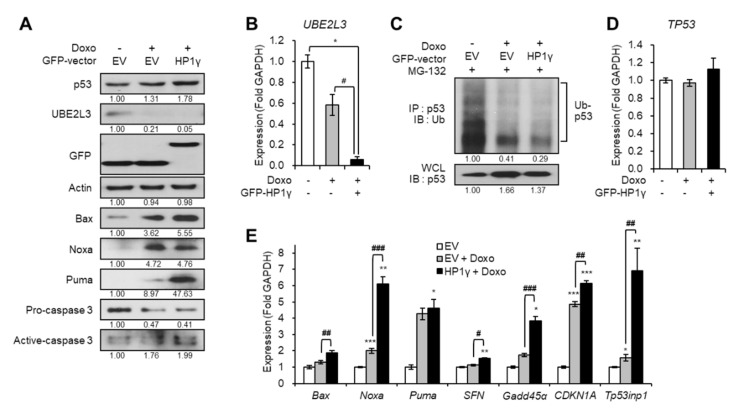
Overexpression of HP1γ boosts DNA damage-mediated p53 elevation. (**A**) Immunoblot analysis of HeLa cells expressing GFP-empty vector or GFP-HP1γ in the presence of doxorubicin (1 μg/mL, 1 h). (**B**) The mRNA levels of *UBE2L3* gene in HeLa cells expressing GFP-empty vector or GFP-HP1γ in the presence of doxorubicin (1 μg/mL, 1 h). (**C**) Immunoblot analysis of p53 immunoprecipitates (IP) and whole cell lysates (WCL) from HeLa cells expressing GFP-empty vector or GFP-HP1γ in the presence of doxorubicin (1 μg/mL, 1 h) after MG-132 treatment. (**D**,**E**) The mRNA levels of *TP53* gene (**D**) and p53 target genes (**E**) in HeLa cells expressing GFP-empty vector or GFP-HP1γ in the presence of doxorubicin (1 μg/mL, 1 h). Data are presented as the mean ± SEM (*n* = 3). * *p* < 0.05; ** *p* < 0.01; *** *p* < 0.001. vs. the control group; # *p* < 0.05, ## *p* < 0.01 and ### *p* < 0.001 vs. doxo-treated group.

**Figure 5 ijms-21-05976-f005:**
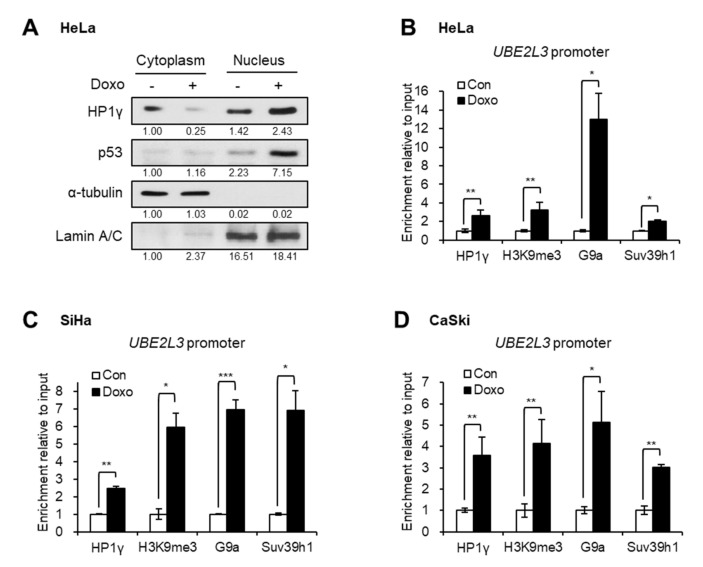
Doxorubicin treatment induces nuclear import and *UBE2L3* promoter binding of HP1γ. (**A**) Immunoblot analysis of cytoplasmic and nuclear extracts from HeLa cells treated with doxorubicin (1 μg/mL, 1 h). (**B**–**D**) HeLa (**B**), SiHa (**C**), and CaSki (**D**) cells were treated with doxorubicin (1 μg/mL, 1 h), followed by ChIP-qPCR analyses for HP1γ, trimethylated H3K9 (H3K9me3), G9a, and Suv39h1 antibodies in the promoter region of *UBE2L3* gene. Data are presented as the mean ± SEM (*n* = 3). * *p* < 0.05; ** *p* < 0.01; *** *p* < 0.001.

**Figure 6 ijms-21-05976-f006:**
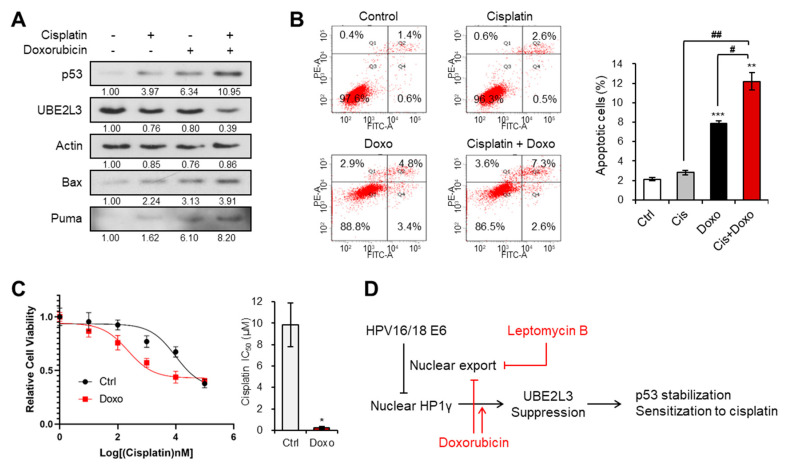
Doxorubicin increases the response of cervical cancer cells to cisplatin. (**A**) Immunoblotting analysis of SiHa cells treated with or without cisplatin (5 μM) and doxorubicin (1 μg/mL) for 24 h. (**B**) Apoptotic cell populations of SiHa cells treated with or without cisplatin (5 μM) and doxorubicin (1 μg/mL) for 24 h were obtained with flow cytometry analysis. (**C**) Viability and IC_50_ values of cisplatin in SiHa cells treated with doxorubicin (1 μg/mL) for 24 h. IC_50_ values of cisplatin in the cells were calculated from the viability graph with GraphPad Prism 8 (*n* = 3). (**D**) Molecular model describing the effects of leptomycin B and doxorubicin on HP1γ-mediated p53 stabilization in cervical cancer cells. Data are presented as the mean ± SEM (*n* = 3). * *p* < 0.05; ** *p* < 0.01; *** *p* < 0.001 vs. the control group; # *p* < 0.05 and ## *p* < 0.01 vs. single agent-treated group.

**Table 1 ijms-21-05976-t001:** Sequences of primers for mRNA used in RT-qPCR.

Gene	Forward	Reverse
*GAPDH*	5′-GAGTCAACGGATTTGGTCGT-3′	5′-TTGATTTTGGAGGGATCTCG-3′
*Bax*	5′-TCTACTTTGCCAGCAAACTGG-3′	5′-TGTCCAGCCCATGATG GTTCT-3′
*Noxa*	5′-AGAGCTGGAAGTCGAGTGT-3′	5′-GCACCT TCACATTCCTCTC-3′
*Puma*	5′-GACCTCAACGCACAGTA-3′	5′-CTAATTGGGCTCCATCT-3′
*SFN*	5′-TTTCCTCT CCAGACTGACAAACTGTT-3′	5′-TAGAACTGAGCTGCAGCTGTAAA -3′
*Gadd45α*	5′-TGCGAGAACGACATCAACAT-3′	5′-TCCCG GCAAAAACAAATAAG-3′
*CDKN1A*	5′-CACCGAGACACCACTGGAGG-3′	5′-GAGAAGATCAGCCGGCGTTT-3′
*Tp53inp1*	5′-TGTTGCAGCTCTTGCTGCTCA-3′	5′-GCTGATGAACAACCCAGCCAT-3′
*TP53*	5′-GAGGGATGTTTGGGAGATGTAAGAAATG-3′	5′-TTCACAGATATGGGCCTTGAAGTTAGAGAA-3′
*UBE2L3*	5′-TTGACCCTTTGTAGGATTGGAATT-3′	5′-CGACCCCAGACTGGTGCTT-3′

**Table 2 ijms-21-05976-t002:** Sequences of primers for mRNA used in ChIP-qPCR.

Gene	Forward	Reverse
*UBE2L3*	5′-GCCAGGCACTGGTTGTAAC-3′	5′-AGTGGGAAGCACACAGTAGG-3′

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
