# Peer review of "HP1γ Sensitizes Cervical Cancer Cells to Cisplatin through the Suppression of UBE2L3"

_ijms, 2020, doi:10.3390/ijms21175976_

Round 1
Reviewer 1 Report
Platinum resistance represents one of the major causes of failure of therapies in oncological gynecology and the final step of these diseases.
The authors below report the explanation of a resistance mechanism and a possibility of improving platinum sensitivity through the use of two drugs Leptomycin B and doxorubicin.
For when the action of leptomycin is not unknown in the literature, especially in HPV-related tumors such as that of the uterine cervix, the article has some peculiarities.
Introduction
Line 33-34
“Early-stage cervical cancer can be cured with surgery and concurrent chemoradiotherapy”however, the treatment options for patients with advanced or recurrent cervical cancer are very limited ;
This statement is not true: early stage can be treated with surgery alone or with surgery and radiotherapy, advanced stages can be treated with exclusive radiochemotherapy (Fanfani et al Eur J Surg Oncol. 2016 and Perrone et al. Cancers (Basel). 2020;).
Better this way: in the early stages surgery represents the primary treatment possibly associated with radiotherapy, in the advanced stages radiochemotherapy or chemotherapy alone are the most appropriate choice although when disease recurs treatments options are very limited.
Line 41
Please, discuss briefly the use of antiangiogenic in the treatment of advanced cervical cancer. This completes the discussion of available therapies and reinforces your hypothesis (Tewari KS. Lancet. 2017)
Line 70-72
As for cisplatin, a brief introduction to the other two drugs would be appropriate. The journal you are writing on is not specific to oncologists
Line 72
Both? LMB and doxorubicin
Line 79-80
Here you talk about results but it is not clear if they are those of this study or others. If they are those of the study so it is not the appropriate place. The aims of the study are missing.
Results
Line 83
The cells are all resistant to platinum but perhaps reiterating it makes the study clearer and removes unnecessary doubts from the reader
Line 204
Figure 6
Can you make a comparison chart between leptomycin B and doxorubicin in order to highlight its effectiveness?
the use of the two drugs together is feasible?
Author Response
Platinum resistance represents one of the major causes of failure of therapies in oncological gynecology and the final step of these diseases.
The authors below report the explanation of a resistance mechanism and a possibility of improving platinum sensitivity through the use of two drugs Leptomycin B and doxorubicin.
For when the action of leptomycin is not unknown in the literature, especially in HPV-related tumors such as that of the uterine cervix, the article has some peculiarities.
We thank the reviewer for recognizing the potential importance of our findings. We feel our responses to his/her criticisms have greatly improved the manuscript, including a number of additional experiments and corrections. Our responses to the reviewer’s queries are described point-by-point below.
<Introduction>
Line 33-34. “Early-stage cervical cancer can be cured with surgery and concurrent chemoradiotherapy” however, the treatment options for patients with advanced or recurrent cervical cancer are very limited ; This statement is not true: early stage can be treated with surgery alone or with surgery and radiotherapy, advanced stages can be treated with exclusive radiochemotherapy (Fanfani et al Eur J Surg Oncol. 2016 and Perrone et al. Cancers (Basel). 2020;). Better this way: in the early stages surgery represents the primary treatment possibly associated with radiotherapy, in the advanced stages radiochemotherapy or chemotherapy alone are the most appropriate choice although when disease recurs treatments options are very limited.
We thank the referee for recommending more proper way to demonstrate the treatment options for cervical cancers patients. As the referee recommended, we corrected the sentence and reference supporting the sentence. Please see page 1, lines 33-36.
Line 41. Please, discuss briefly the use of antiangiogenic in the treatment of advanced cervical cancer. This completes the discussion of available therapies and reinforces your hypothesis (Tewari KS. Lancet. 2017).
As the reviewer suggested, we demonstrated the use of anti-angiogenesis therapy for advanced cervical cancer (page 1, lines 42-43). Accordingly, the reference 4 was replaced with more recent one, which was stated by the reviewer.
Line 70-72. As for cisplatin, a brief introduction to the other two drugs would be appropriate. The journal you are writing on is not specific to oncologists
According to the comment of reviewer 2, we updated the last paragraph of introduction part to highlight the aim of this study rather than the summary of results. Additionally, we briefly demonstrated leptomycin B and doxorubicin in that part (page 2, lines 76-77), with newly added reference (ref 16). Besides, the mechanisms of action of the two drugs, which could not be explained sufficiently in the introduction part, are demonstrated in detail in the discussion part (page 9, lines 253-257; page 9, lines 273-276).
Line 72. Both? LMB and doxorubicin
We corrected the introduction part including the confusing sentence that the reviewer pointed out.
Line 79-80. Here you talk about results but it is not clear if they are those of this study or others. If they are those of the study so it is not the appropriate place. The aims of the study are missing.
This point was also raised by reviewer 2. According to the reviewer’s opinion, we updated the last paragraph of introduction part to include the aims of our study and clarify new findings of our study.
<Results>
Line 83. The cells are all resistant to platinum but perhaps reiterating it makes the study clearer and removes unnecessary doubts from the reader
We agree with the reviewer’s opinion that stating the sensitivity to cisplatin of three cell lines would clarify our study. Therefore, we added the statement about it at page 2, line 87-88 of revised manuscript. Moreover, we indicated the IC50 values for HeLa, SiHa, and CaSki cells (page 2, line 93).
Line 204. Figure 6. Can you make a comparison chart between leptomycin B and doxorubicin in order to highlight its effectiveness? the use of the two drugs together is feasible?
First, we apologize for the mislabeling in Figure 6. The FACS data displayed in Figure 6B indicate the effects of doxorubicin, not LMB.
As the reviewer requested, we here suggest comparison chart of the apoptotic cell populations between leptomycin B and doxorubicin (please see the graph in right side). This data show that the effectiveness of doxorubicin and leptomycin B on apoptosis is not significantly different.
However, as previously reported (PLoS One 2012; 7(3): e32895) the timing of activity is different between the two drugs. Upon combined treatment of doxorubicin and leptomycin B, doxorubicin promotes initial activation of p53 and consequently CRM1 function blocking by leptomycin B to accumulate activated p53 in the nucleus (PLoS One 2012; 7(3): e32895). Thus, it seems that the use of doxorubicin and leptomycin B together would induce synergistic effects, but our study address the effects of leptomycin B and doxorubicin on sensitizing cervical cancer cells to cisplatin, not the co-treatment of leptomycin B and doxorubicin. So, we ask the reviewer and editor to let this story be covered in another study, rather than in this study.

Reviewer 2 Report
-The last paragraph of the introduction is summarising the results of the study. The authors are encouraged to end the introduction with the aim of the study rather than summary of results. - For an easier understanding of the text, it is recommended that in the results section, the type of cells based on sensitivity or resistance to cisplatin be mentioned. - Result 1, Figure 1: Please include the IC50 values for each cell line, with/out HP1 over expression. -IC50 values are measured based on cell viability. Please indicate the seeding density of the cells and the plates that were used for the experiments. Do you mean 6-well plates? - What positive and negative controls did you use to perform annexin V/PI staining? Please indicate them in the text. - How did you come up with the 5uM concentration of cisplatin for the experiments?Author Response
-The last paragraph of the introduction is summarising the results of the study. The authors are encouraged to end the introduction with the aim of the study rather than summary of results.
We have updated the last paragraph of the introduction to highlight the aim of our study. Please see the revised manuscript (page 2, lines 73-84).
- For an easier understanding of the text, it is recommended that in the results section, the type of cells based on sensitivity or resistance to cisplatin be mentioned.
We agree with the reviewer’s opinion that distinguishing the type of cells based on sensitivity/resistance to cisplatin would clarify our study. In the revised manuscript (page 2, lines 94-95), we added the different activity of p53/p21 signaling in cisplatin-sensitive HeLa cells and cisplatin-resistant SiHa and CaSki cells, which can give more rationale for our observation of p53 level.
- Result 1, Figure 1: Please include the IC50 values for each cell line, with/out HP1 over expression.
As the reviewer recommended, we indicated the IC50 values for HeLa, SiHa, and CaSki cells expressing HP1γ or not (page 2, lines 93).
-IC50 values are measured based on cell viability. Please indicate the seeding density of the cells and the plates that were used for the experiments. Do you mean 6-well plates?
We used 6-well plates and seeded the cells at 105 cells/well. 24 h after seeding, the cells were treated with chemicals, such as cisplatin, LMB, or doxorubicin, and then the cell number was counted every 8 hours. We added this detailed method at page 10, lines 315-316 of revised manuscript.
- What positive and negative controls did you use to perform annexin V/PI staining? Please indicate them in the text.
First, we apologize for the mislabeling in Figure 6. The FACS data displayed in Figure 6B indicate the effects of doxorubicin, not LMB. As regards positive and negative controls, we used DMSO as a negative control and doxorubicin and leptomycin B could serve as positive controls, because it has already been well known that the two drugs cause apoptosis (Oncogene 2005, 24, 2765-4777; Cancer Res. 2009, 69, 510-517). As the reviewer suggested, we included this information to Materials and Methods section (page 10, lines 325-327)
- How did you come up with the 5uM concentration of cisplatin for the experiments?
We selected the treatment dose of cisplatin (5 μM) based on IC50 values displayed in Figure 1B. IC50 value of HeLa cells was markedly lower than 5 μM, while IC50 value of SiHa cells was higher than 5 μM and that of CaSki cells was similar to 5 μM. Thus, treatment of each cell with 5 μM of cisplatin induced severe cell death in HeLa cells, mild response in CaSki cells, and had little effects on SiHa cells (Figure 1A and 1C). Hence, we thought that 5 μM of cisplatin could lead to different degrees of reactions from the three cell lines.
Round 2
Reviewer 1 Report
Thanks for the answers I am satisfied. Nothing else to add. Congratulations to the authors.